# Tit for Tat: Abusive Supervision and Knowledge Hiding-The Role of Psychological Contract Breach and Psychological Ownership

**DOI:** 10.3390/ijerph17041240

**Published:** 2020-02-14

**Authors:** Usman Ghani, Timothy Teo, Yan Li, Muhammad Usman, Zia Ul Islam, Habib Gul, Rana Muhammad Naeem, Humera Bahadar, Jing Yuan, Xuesong Zhai

**Affiliations:** 1College of Education, Zhejiang University, Hangzhou 310000, China; gusman@mail.ustc.edu.cn (U.G.); yanli@zju.edu.cn (Y.L.); 2School of Education, Murdoch University, Murdoch 6150, Western Australia; Timothy.Teo@murdoch.edu.au; 3Department of Management Sciences, Preston University, Kohat 26000, Pakistan; muhammad.usman@preston.edu.pk; 4School of Management, University of Science and Technology of China, Hefei 230026, China; ziazadian83@mail.ustc.edu.cn (Z.U.I.); naeemrana426@gmail.com (R.M.N.); 5Business School, University of International Business and Economics, Beijing 100875, China; habibuibe2020@gmail.com; 6Department of Management Sciences, Hazara University, Mansehra 21120, Pakistan; humayousafzai85@gmail.com; 7School of Foreign Study, Anhui Sanlian University, Hefei 230026, China; jing.jane.yuan@gmail.com

**Keywords:** knowledge hiding, abusive supervision, psychological contract breach, psychological ownership

## Abstract

The extant literature has focused on individuals’ knowledge-sharing behavior and its driving factors, which stimulate the knowledge transmission and exchange in organizations. However, little research has focused on factors that inhibit knowledge sharing and encourage individuals to hide their knowledge. Therefore, based on social exchange and displaced aggression theories, the study proposed and checked a model that examined the effect of abusive supervision on knowledge hiding (KH) via a psychological contract breach (PCB). The Psychological ownership was regarded as a boundary condition on abusive supervision and KH relationship. Using a time-lagged method, we recruited 344 full-time employees enrolled in an executive development program in a large university in China. The findings show that PCB mediates the association between abusive supervision and KH. Similarly, psychological ownership moderates the association between abusive supervision and KH. Employees with high psychological ownership minimized the effect of abusive supervision on KH. Based on study findings, contributions to theory and practice, limitations, and future directions are discussed.

## 1. Introduction

Knowledge management is a process to ensure the transmission of knowledge throughout the organization [1,2,3]. Organizational development and growth depend on how individuals share knowledge collectively in the workplace to achieve organizational goals [4,5]. However, although knowledge-sharing among employees adds values to firms, employees become unwilling, reluctant, and dispassionate to share knowledge with their colleagues, which are known as knowledge hiding (KH) [6,7]. The reasons for hiding knowledge is to bear personal cost by sharing loss of status, power, ownership, and unrecognizability [8,9,10], which results in an irrelevant exchange of knowledge possessed by individuals. In other words, individuals engage in KH, which is defined as an “intentional attempt by an individual to withhold or conceal knowledge that has been requested by another person” [6] (p. 65).

Individuals’ behaviors of hiding knowledge push them to avoid knowledge-sharing activities [11]. Prior studies have identified three major factors influencing KH including individual factors, knowledge content, and organizational factors. For example, research has established the relationship between personal level factors and KH, such as the perceived value of knowledge, psychological entitlement, professional commitment, and psychological ownership of knowledge [12,13,14,15]. Similarly, knowledge-related factors that contribute to KH include task relatedness and complexity of knowledge [6]. Previous research has also explored various organizational factors in relation to KH, including leadership styles, culture, knowledge management system, policies, goal orientation, and politics [16,17,18,19,20]. However, the abusive behaviors of supervisors, which lead to KH behaviors, is not explained fully. Khalid et al. [18] investigated how abusive behaviors of supervisors result in employees’ knowledge-hiding behavior through interpersonal justice.

Abusive supervision is defined as “subordinates’ perceptions of the extent to which supervisors engage in the sustained display of hostile verbal and nonverbal behaviors, excluding physical contact” [21] (p. 178). Supervisors’ abusive behaviors, such as criticizing, ridiculing, and intimidating subordinates in front of others, undermine subordinates’ perceptions of the psychological contract [22]. Given that a psychological contract breach (PCB) leads to counterproductive behaviors, i.e., abuse, withdrawal, and deviant behaviors [23,24,25], we, therefore, expect that employees would tend to engage in KH behaviors when their psychological contract is breached. Notably, the literature on abusive supervision in relationship with KH is lacking. In particular, prior studies have not noted the important mechanism of PCB in the relationship between abusive supervision and KH. Moreover, previous research [26] suggested that the detrimental outcomes of abusive supervision may be mitigated by individual characteristics. Psychological ownership is defined by Pierce, Kostova, and Dirks [27] as the state in which employees sense that the target of ownership or a piece of that particular target is theirs. Importantly, previous research has been unclear on the possible mitigating role of psychological ownership in the abusive supervision-KH relationship. Therefore, we further posit that the positive association between abusive supervision and KH are more likely to be weakened by individuals having high ownership perceptions (see Figure 1).

Drawing on social exchange and displaced aggression theories, our study has several contributions to the literature. First, we extend the literature by examining the predictor (i.e., abusive supervision) for KH in a new context. We also posit that PCB is an important underlying mechanism between abusive supervision and KH behavior. We argue that employees who felt abused by their supervisors will feel unobligated to benefit their organizations and will hide knowledge, which may decrease organizational effectiveness. Furthermore, prior studies have shown that psychological ownership is an important predictor of individuals’ attitudes and behaviors [28,29]. Hence, it can be used as a significant factor for attenuating the relationship between abusive supervision and KH.

In sum, the current study addresses three important research questions: (1) Does abusive supervision positively influence individuals’ KH behaviors? (2) Does PCB mediate the relationship between abusive supervision and KH? (3) Does psychological ownership moderate the relationship between abusive supervision and KH?

## 2. Theoretical Background and Hypothesis Development

### 2.1. Abusive Supervision and Knowledge Hiding

Abusive behaviors of supervisors include mocking, rudeness, public ridicule, belittlement, and other hostile behaviors [15], such as employing the silent treatment, breaking promises [30], and aggressive eye contact [31]. Past studies have shown that abusive behaviors of supervisors are associated with employees’ undesirable attitudinal and behavioral outcomes, such as low job performance [30], a decrease in organizational citizenship behavior [32], an increase in deviant workplace behavior, job burnout, and emotional exhaustion [33,34,35,36,37]. Moreover, prior studies tested several individual (i.e., professional commitment and psychological entitlement) and organizational factors (i.e., organizational culture, policies, leadership styles) as antecedents with KH [12,13,14,15,16,17,18,20]. However, one such undesirable behavioral outcome is KH, which is largely ignored in the abusive supervision literature.

Knowledge hiding is the intentional attempt of an individual to conceal knowledge that has been requested by another individual [6]. KH behaviors consisted of three dimensions, such as rationalized hiding, playing dumb, and evasive hiding [6]. In rationalized hiding, an individual is trying to give justification to the knowledge seeker or blame a third party for not providing the requested knowledge. Similarly, in playing dumb, an individual expresses himself as ignorant of the knowledge being requested by the knowledge seeker. Evasive hiding is defined as the hider providing inappropriate facts or promising to provide information in the future to mislead the knowledge seeker with no such real intentions.

Based on the social exchange theory (SET), social relationships are based on reciprocal benefits [38,39]. Consistent with the norm of reciprocity [40], the attitudes and behaviors of subordinates are linked with supervisors’ behaviors. When subordinates receive respectful treatment from the supervisors, they tend to return that with positive attitudes and behaviors. In contrast, when subordinates are treated in an abusive fashion, they are inclined to reciprocate with counterproductive behaviors [41]. In a similar vein, displaced aggression theory (DAT) [42] also suggests that individuals victimized by abusive leaders do not express their negative attitude and behaviors directly toward their supervisors because of their power and authority [43]. Instead, they show their aggression toward convenient targets or victims, such as their colleagues or coworkers [44,45]. One such retaliatory reaction directed toward colleagues/coworkers may be KH behavior. On the basis of previously mentioned literature and theoretical reasoning, it is expected that abusive supervision may be positively associated with KH behavior. Therefore, we hypothesize that:

**Hypothesis** **1.** **(H1).**Abusive supervision positively affects knowledge hiding.

### 2.2. Mediating Role of a Psychological Contract Breach

The psychological contract is defined as “individual beliefs, shaped by the organization, regarding terms of an exchange agreement between individuals and their organization” [46]. PCB has been defined as the cognitive perception that an employee has not received everything that was formally or informally promised by the organization [47]. PCB occurs when the organization or its representatives, such as supervisors, fail to meet the expectations of their employees [22,47]. Abusive supervisors often mock employees, belittle and degrade them in front of others, make a joke out of them, and display other hostile behaviors, which makes subordinates feel disrespected and their expectations unfulfilled. Thus, subordinates under abusive supervisors tend to experience PCB.

PCB is related to a range of undesirable employee attitudes and behaviors. For example, a psychological contract breach is negatively associated with an employee’s trust in management [48], job satisfaction [49], intentions to stay with the organization [50], employee performance [51], citizenship behaviors [50], civic virtue behaviors [52], and employee commitment [49,53]. Furthermore, some other scholars concluded that PCB is positively associated with workplace deviant behaviors [54], neglect of job responsibilities [55], job burnout [56], employee cynicism [57], higher absenteeism [48], and revenge cognitions [54].

The relationship between supervisory abuse, PCB, and knowledge hiding can be explained through the SET lens. The notion of negative reciprocity is based on SET, which suggests that, when personnel perceives supervisory abuse, they are more likely to experience the breach of the psychological contract, and, consequently, they are motivated to react in a negative way [58]. Hence, abused employees will likely experience a breach of psychological contract and, consequently, be motivated to exhibit counterproductive work behavior (e.g., KH behavior) in a deliberate way to balance the exchange relationship. Based on the above literature and theoretical argumentation, we presume the following hypothesis:

**Hypothesis** **2.** **(H2).**Psychological contract breach mediates the relationship between abusive supervision and knowledge hiding.

### 2.3. Moderating Role of Psychological Ownership

Prior studies suggest that subordinate’s reaction toward abusive supervision may vary in nature and severity. For example, Tepper [26] suggested that, despite the detrimental outcomes of abusive supervision, not every individual will perceive the mistreatment from a supervisor as abusive. The relationship between abusive supervision and various employees’ outcomes may be mitigated by individual characteristics that may lessen the effect of adverse reactions to abusive supervision. Psychological ownership with the organization is one such characteristic. Pierce et al. [27] defined psychological ownership as “the state in which individuals feel as though the target of ownership or a piece of that target is theirs” (p. 86). The construct of psychological ownership has gained scholarly attention in today’s dynamic knowledge-based economy [41,59]. Avey, Avolio, Crossley, and Luthans [60] described psychological ownership as a multidimensional construct, which is illustrated as a sense of belonging, self-efficacy, accountability, and identity. Individuals with psychological ownership have a greater sense of belonging to the target, feel more efficacious and more accountable, and have a greater sense of personal identification [61].

In addition, the psychological ownership concept is a significant predictor of attitudes, behaviors, and motives at the individual-level in a workplace [28,29]. Avey, Avolio, Crossley, and Luthans [62] investigated psychological ownership and found a positive impact on citizenship behavior, job satisfaction, and commitment, while a negative relation is found with the deviant behavior of employees. We believe that individuals with greater psychological ownership will react more constructively than those who have less psychological ownership when they experience abusive supervision. This is because subordinates with greater psychological ownership are inclined toward the target, and, hence, they are considered to be deliberate and thoughtful in their reaction to workplace stressors. Given that, abusive supervision is one of the destructive workplace stressors [63]. Therefore, we believe that psychological ownership will mitigate the reaction of employees to abusive supervision as KH. Thus, our proposed hypothesis is:

**Hypothesis** **3.** **(H3).**Psychological ownership moderates the relationship between abusive supervision and knowledge hiding such as when psychological ownership is high. This relationship will be weaker and vice versa.

## 3. Methods

### 3.1. Participants and Procedure

The study was approved by the School of Educational Technology, Beijing Normal University’s review board, and it included detailed consent procedures. The participants were informed that the study would assess perceived abusive supervision, psychological contract breach, psychological ownership, and knowledge hiding. All the participants signed a written consent form, and all the procedures were carried out in accordance with relevant regulations.

In this study, data were gathered from full-time employees enrolled (only weekend days) in an executive development program at large universities in China. The respondents were sent by their organization as a part of their career development program. A diverse pool of 344 respondents was included in the manufacturing and services sectors. According to Highhouse [64], the diversified sample of different functional groups and organizations increase the generalizability of the research findings. Our sample consisted of those respondents who had at least one year working experience with their immediate boss in their organization. All the respondents were very promising when it comes to participating in the survey. The scale used in the study was in the Chinese language because we collected data in the Chinese context and from Chinese respondents. For this purpose, the questionnaire was translated into Chinese through the back-translation method, which was recommended by Brislin [65]. We translated the English version into the Chinese language with the help of two bilingual Chinese professors, who were proficient in both languages. Then the Chinese version of the instrument was translated into English to achieve uniformity, with the help of another two professors who had professional experience of translation. Then, a pilot study was conducted with 45 respondents before the primary study to check the content accuracy of items and understandability of the questions. The participants were asked not only to fill the questionnaire but also to provide feedback about the items’ relevance and completeness to ensure the quality of measurement items. The respondents reported no issues or confusion about the survey.

The data were collected in a different range of times to minimize the common method variance issue [66]. A time lag of three weeks is considered between distributing questionnaires. A unique identifier is assigned to each participant response to match time 1 (T1), time 2 (T2), and time 3 (T3) surveys. The unique identifier included only a numerical number to match T1, T2, and T3 responses and was not included with any personal information to ensure respondent anonymity. At T1, the respondent filled set one of the questionnaires, including demographics and independent variable (abusive supervision) items. Next, at T2, the second set of questionnaires were filled, including moderation (psychological capital) and mediation (psychological contract breach) variable items. Lastly, at T3, the third set of the questionnaire was filled, including the dependent variable (knowledge hiding) items. A total of 355 questionnaires were distributed in which eleven questionnaires were uncompleted and excluded. The response rate was 96.9%. To carry on analysis, we used the final sample (*N* = 344). The respondents’ demographics are given in Table 1.

### 3.2. Measures

All the constructs were adapted from the previous studies. The constructs items were ranked on a 5-point Likert scale ranging from 5 (strongly agree) to 1 (strongly disagree).

#### 3.2.1. Abusive Supervision

Abusive supervision was defined as the perception of subordinates to which extent their supervisors engaged in persistently showing hostile behavior: verbal and nonverbal, without any physical contact, and measured with a 15-items scale (α = 0.90) adapted from the study of Tepper [21]. The sample items of the scale are “My boss/supervisor puts me down in front of others.” and “My boss/supervisor ridicules me.”

#### 3.2.2. Psychological Ownership

Psychological ownership was defined as the feeling of possession and ownership about the organization or a piece of that organization and measured with a 7-items scale (α = 0.87) adapted from the study of Van Dyne and Pierce [67]. The sample items of the scale are “This is MY organization.” and “I sense that this organization is OUR company.”

#### 3.2.3. Psychological Contract Breach

The psychological contract breach is defined as individuals’ perceptions that their supervisor has failed to meet all or any of the obligations owed to them. This breach is measured using a five-item scale of a psychological contract breach (α = 0.92) adapted from the study of Robinson and Morrison [68]. The sample items included are “My employer/supervisor has broken many of its promises to me even though I have upheld my side of the deal.” and “I have not received everything promised to me in exchange for my contributions.”

#### 3.2.4. Knowledge Hiding

Knowledge hiding is defined as an individual’s intentional attempt to conceal knowledge requested by a knowledge seeker. The scale was adapted from the study of Peng [14] (α = 0.91) and measures knowledge hiding using a five-item scale. The sample items include “I withhold helpful information or knowledge from others.” and “Do not transform personal knowledge and experience into organizational knowledge.”

### 3.3. Analytical Approach

In this study, several statistical techniques were employed using SPSS and AMOS for data analysis. Factor loadings, Cronbach’s alpha, and composite reliability were calculated, and the discriminant validity evaluated using the average variance extracted (AVE) to measure the reliability and validity of the constructs. Confirmatory factor analysis (CFA) analysis was conducted to determine the model fitness indices, such as χ^2^ (CMIN/df), root mean square error of approximation (RMSEA), incremental fit index (IFI), Tucker-Lewis index (TLI), and comparative fit index (CFI). We utilized Hayes [69] PROCESS macro to estimate both mediation and moderation effects. We conducted a bootstrap analysis using path analytic procedures to test our hypotheses and to assess the significance of the indirect effects [70,71,72]. Bootstrap is considered a more sophisticated and reliable method for estimating indirect effects in the social sciences [69,73]. In addition, to proceed toward our main analysis, we checked the data for wrong coding, missing values, outliers, and multicollinearity. We did not observe such issues in our dataset.

## 4. Results

### 4.1. Measurement Tests

A common method bias (CMB) was assessed by following the approach of Podsakoff et al. [66]. First, CMB was checked using confirmatory factor analysis (CFA). The original measurement model was linked with a common latent factor (CLF) and the outcome did not show any significant loss in the factor loadings. Second, the Harman single-factor test was applied and the results showed the total variance explained, which was accounted by one factor that was 34%, which is less than the threshold value of 50% [74]. Hence, the CMB was not an issue in this study. CMB exists when the study variables have higher inter-correlations (r > 0.90) among them [75]. The correlation results indicate that the higher inter-correlations were not found among the study variables. These statistics indicated that a common method bias was not a severe issue in the data.

Constructs reliability and validity were extracted through Cronbach’s alpha, composite reliability (CR), average variance extracted (AVE), and factor loadings of the items [76,77]. Table 2 shows that the Cronbach’s alpha, CR, and value of AVE for each construct achieved the threshold value of 0.70, 0.60, and 0.50, respectively. Convergent validity was extracted through the value of factor loadings of each construct. The factor loadings of each item were higher than the cut-off standard of 0.70 (see Table 2), which indicates that all constructs exhibit excellent convergent validity. Discriminant validity was checked by comparing the square root of the AVE and the inter-correlation of each construct. According to Fornell and Larcker [76], the square root value of AVE for each measure should be larger than the value of each inter-correlation measure. Table 3 indicates that all constructs’ square root values of AVE are larger than all the correlations among all the constructs, which validates the discriminant validity of the model.

A CFA was run in AMOS 21 to assess the suggested measurement model. We linked the study variables in a multidimensional manner and the findings depicted that the model is a good fit, acceptable, and meets the threshold values suggested by Hair et al. [77] and Hu and Bentler [78]. Table 2 shows all items are well-loaded upon their corresponding constructs and the value of the estimates for all constructs is significant. The model fit indices indicate good fit to the data: χ^2^ (CMIN/df) = 2.89, *p* < 0.000, CFI = 0.913, TLI =0.906, IFI = 0.914, RMSEA = 0.074, and SRMR = 0.041. Thus, our measurement model is acceptable. We then conduct further analysis to test our proposed hypotheses using PROCESS macro [69] in SPSS 22.

### 4.2. Hypothesis Testing

Table 3 indicates the correlations among the study variables. All correlation coefficients of the study variable were in the proposed direction.

We run PROCESS macro suggested by Hayes [69] in SPSS to find a direct and indirect effect and test H1 and H2. Table 4 shows the results of the mediation effect of PCB. Abusive supervision has a positive significant effect on knowledge hiding as β = 0.25** and *t* = 4.20 and, thus, H1 is accepted. The psychological contract breach mediates the relationship between abusive supervision and knowledge hiding because the CI (0.0271, 0.1476) for an indirect effect did not include zero and, thus, H2 is accepted.

Table 5 indicates the results of the moderating role of psychological ownership between abusive supervision and knowledge hiding. The significant interaction (abusive supervision x psychological ownership) value (β = −0.24**, *t* = −0.253, *p* < 0.01) supports H3. Psychological ownership was split into low (–1 SD) and high (+1 SD) levels to investigate the nature of interaction effects. The graphical illustrations of the moderating effects of psychological ownership are shown in Figure 2. The positive relationship between abusive supervision and knowledge hiding is weaker (β = 0.01, t = 0.09, CI = −0.1751 to 0.1916) at high levels of psychological ownership. The same relationship is positive and stronger (β = 0.37**, *t* = 3.87, CI = 0.1854 to 0.5691) at low levels of psychological ownership. Thus, these findings provided further support for the moderation hypothesis.

## 5. Discussion

The goal of this study was to determine whether supervisory abuse leads individuals to knowledge-hiding behaviors. In parallel to prior studies [18], our results show that individuals are more prone to KH behaviors in the presence of abusive leadership. In addition, the study findings revealed that PCB mediates the relationship between abusive supervision and KH. More precisely, the results indicate that when employees perceive the abusive behavior of their supervisor, they infer a breach in their psychological contract and, subsequently, engage in KH behaviors. The study also concludes that, as a boundary condition, psychological ownership weakens the positive relationship between abusive supervision and KH. It means individuals with high psychological ownership show their alignment with organizational goals and invest their resources in acting as though the organization is theirs and are more committed even in the presence of abusive leadership. Consequently, the individual’s feelings of ownership buffer the negative effect of a supervisor’s abusive behaviors on employees’ retaliatory reactions toward their organizations such as hiding their knowledge.

## 6. Implication

### 6.1. Theoretical Implications

Our study contributes significant implications to the theory. The current study integrated research on displaced aggression and social exchange theories, which provides empirical support to fill an important gap in our understanding of how the relationship between supervisory abuse and KH is mediated through a psychological contract breach and how individual characteristics, such as psychological ownership, moderate the relationship between abusive supervision and knowledge hiding. Based on SET and displaced aggression theory in parallel with a prior study [18], our result shows that supervisory abuse is positively related to employees’ KH behaviors in a new cultural context. These findings suggest that the supervisor abusive behaviors compel individuals to respond in a negative way. It is because individuals think their knowledge base to be valuable and the sense of ill-treatment or disregard will force them toward knowledge hiding behaviors [79]. Psychological contract breach, which refers to individuals’ perceptions that their supervisor has failed to meet all or any of the obligations owed to them [47], has been proposed as an important psychological mechanism that leads to certain undesired organizational outcomes [24,25]. An individual’s perception of supervisory abuse enables them to develop implicit and explicit beliefs that the organization or its agents (supervisor) break the exchange relationship. As a result and in line with prior studies, our findings suggest that individuals tend to react in a negative reciprocity by engaging in counterproductive work behaviors, such as KH [24], because it is easy for them to control the flow of knowledge from their colleagues as a retaliation.

Moreover, researchers [26,80] have long called for studying more moderators of abusive supervision. Adding to this line of research, the findings of the present study conclude that an individual’s traits, such as psychological ownership, moderates the relationship between supervisory abuse and knowledge-hiding behaviors. In line with the Tepper [26], who stated that the detrimental outcomes of abusive supervision may be mitigated by individual characteristics, our findings confirm that individuals with high psychological ownership perceptions weaken the relationship between abusive supervision and KH and vice versa. We extended this line of research by confirming that psychological ownership decreases the harmful outcome (i.e., KH) of abusive supervision. This result ascertains the theoretical generalizability of studies on abusive supervision beyond the Western context in which it has been largely tested. Moreover, in an Asian culture, where power distance is much higher and abusive behaviors are frequently exhibited in the workplace, one could argue of the necessity of addressing abusive supervision and its mitigating factors as compared to the Western context.

### 6.2. Practical Implications

This study has several managerial implications. First, supervisors’ abusive behaviors are pernicious for the organization’s financial performance [21,81]. KH behaviors inhibit the exchange of information and reduce the organizations’ ability to fetch innovation, which, in turn, poses a threat to the strategic goals of the organization. The higher-ups in the organization may control KH behaviors by discouraging abusive supervisors. One way to reduce abusive supervision is to minimize its dispositional and contextual causes [82,83].

Second, it is empirically evident that supervisory abuse prevails in the workplace [84], and its detrimental consequences have been acknowledged in the extant literature. Therefore, organizations need to initiate certain programs for their leadership to minimize KH behaviors. For example, supervisors may be urged to participate in specialized training with a focus on anger management and interpersonal skills development [30,32]. In addition, organizations can attempt to minimize the incidences of abusive supervision through human resource practices, such as selection and training.

Third, because of the evidence that managers use abuse as a strategic tool [85], abolishing abusive supervision completely is often difficult. As confirmed in our study, psychological ownership provides a buffer against the outcomes of abusive supervision. Therefore, organizations can instill a sense of psychological ownership in the employees. Recently, Dawkins, Tian, Newman, and Martin [86] reasoned that psychological ownership could be improved by bringing structural changes in the organization. This is important because studies indicate that individual differences may affect individuals’ KH differently [87,88].

Fourth, organizations should be attentive to the causes of KH behaviors among their employees. The exchange of knowledge is more likely to occur when employees’ psychological contract with the organization are sustained, which can make employees more obligated to share rather than hide the knowledge. Additionally, organizations should inquire directly or indirectly with the employees on their changing expectations, which would clarify the psychological contract of employees. Organizations should also manifest positive leadership approaches (e.g., servant leadership) in their training programs, which could contribute to the maintenance of the psychological contract of employees with the organization and inhibit their negative behaviors [25].

## 7. Limitations and Future Directions

Our findings should be interpreted by considering several limitations that need to be addressed by future studies. First, even though the present study considers time-lagged design spanning over three periods, we cannot claim causality because it is not purely longitudinal. Therefore, we invite future scholars to adopt longitudinal or experimental studies to allow for stronger causal conclusions [89]. Second, a one-dimensional construct of KH was used in the study. In future studies, a three-dimensional construct of KH behaviors developed by Connelly et al. [6] could be used to gain a more profound understanding of which type of KH is more sensitive to contextual and personal factors. Third, the generalizability of the results is limited because the sample was limited to Chinese employees only. Hence, this model could be tested in the Western context because the Western and Chinese cultural contexts are significantly different, which could interfere with the personnel’s mental and behavioral responses [90]. Fourth, this study explores the KH at an individual level. However, KH at the team-level has not been well documented in the literature [91]. Scholars are encouraged to conduct future studies at the team level because it is unclear if team-level KH is different from individual KH in terms of antecedents and consequences.

## 8. Conclusions

The findings of the current research are quite promising because it provides further quantitative empirical support to the potential role of abusive supervision in relation to the hiding of knowledge. At the same time, our research contributes to the literature by testing an underlying mechanism, i.e., psychological contract breach in the abusive supervision and knowledge-hiding behaviors. This study examined the individual psychological ownership as a boundary condition for abusive supervision-knowledge hiding. Ultimately, our study provides insights that organizations can use to address the problem of supervisory abuse and its deleterious consequences. At the same time, the study sets the stage for further research to understand how other individual and contractual factors may influence individual knowledge-hiding behaviors.

## Figures and Tables

**Figure 1 ijerph-17-01240-f001:**
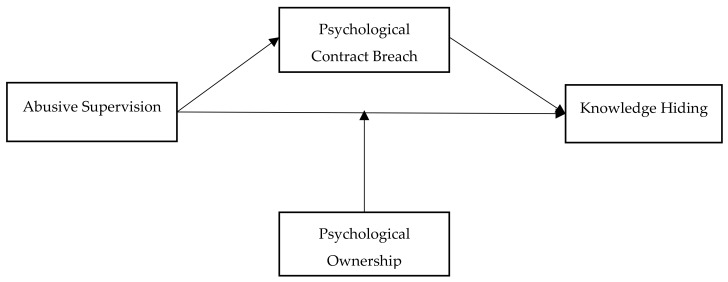
Proposed framework.

**Figure 2 ijerph-17-01240-f002:**
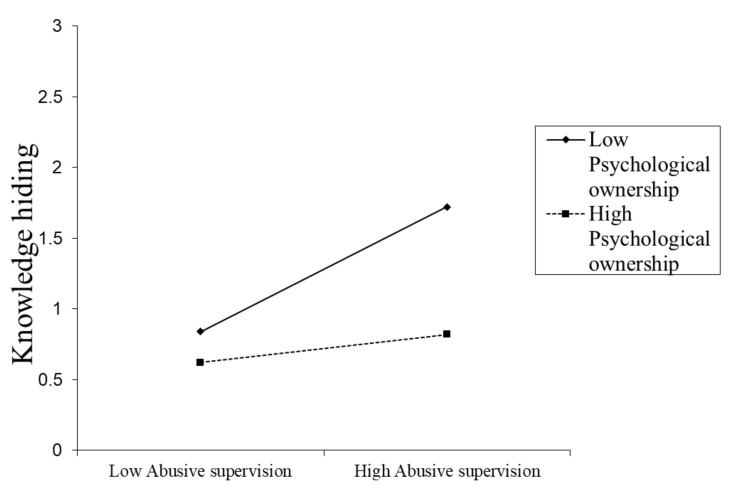
Moderation effect of psychological ownership on abusive supervision and a knowledge hiding relationship.

**Table 1 ijerph-17-01240-t001:** Respondents’ information.

Variables	Categories	Number	Percentage
Gender	Male	180	52.3
	Female	164	47.7
Age (in years)	<30	99	28.8
	31–40	124	36.0
	>50	121	35.2
Education	Master	110	32.0
	Bachelor	140	40.7
	Others	94	27.3
Work experience (in years)	1–5	64	18.6
	6–10	102	29.7
	11–15	144	41.9
	>15	34	9.9

**Table 2 ijerph-17-01240-t002:** Assessment of the measurement model.

Constructs	Factor Loadings	Cronbach’s Alpha	CR	AVE
Abusive Supervision	0.741	0.969	0.969	0.678
	0.735			
	0.795			
	0.881			
	0.861			
	0.881			
	0.737			
	0.838			
	0.846			
	0.844			
	0.882			
	0.885			
	0.884			
	0.794			
	0.715			

Psychological Ownership	0.833	0.919	0.921	0.627
	0.765			
	0.838			
	0.720			
	0.721			
	0.799			
	0.856			

Psychological Contract Breach	0.717	0.922	0.923	0.708
	0.942			
	0.910			
	0.754			
	0.860			

Knowledge Hiding	0.737	0.915	0.916	0.687
	0.855			
	0.852			
	0.875			
	0.818			

**Table 3 ijerph-17-01240-t003:** Descriptive results.

	Mean	SD	1	2	3	4	5	6	7	8
1. Age	2.06	0.79	-							
2. Gender	1.48	0.50	-	-						
3. Education	1.95	0.77	-	-	-					
4. Work Experience	2.28	1.07	-	-	-	-				
5. ABS	3.94	0.91	0.01	−0.01	0.06	0.02	**(0.825)**			
6. PO	2.97	0.76	0.05	0.06	0.01	−0.06	−0.14^**^	**(0.794)**		
7. PCB	3.82	0.93	0.09	−0.03	0.05	−0.06	0.16^**^	−0.02	**(0.843)**	
8. KH	3.73	1.05	0.07	−0.05	0.08	0.01	0.23^**^	−0.24^**^	0.49^**^	**(0.831)**

**Note:** ** *p* < 0.01, ABS = Abusive supervision. PO = Psychological ownership. PCB = Psychological contract breach. KH = Knowledge hiding. The bold values presented in parentheses indicates discriminant validity.

**Table 4 ijerph-17-01240-t004:** Mediation effect.

Outcome: Knowledge Hiding	β	SE	*t*	R^2^
				24
Constant	2.56**	0.38	6.79	
Abusive supervision	0.25**	0.06	4.20	
Age	0.04	0.08	0.52	
Gender	−0.05	0.12	−0.43	
Education	0.06	0.08	0.75	
Work experience	0.01	0.05	0.15	

Outcome: Psychological Contract Breach	β	SE	*t*	R^2^
				0.12
Constant	3.08**	0.34	9.10	
Abusive supervision	0.16**	0.05	2.91	
Age	0.10	0.08	1.39	
Gender	0.01	0.10	0.10	
Education	−0.01	0.08	−0.05	
Work Experience	−0.05	0.04	−1.18	

Outcome: Knowledge Hiding	β	SE	*t*	R^2^
				0.27
Constant	0.93*	0.37	2.52	
Psychological contract breach	0.53**	0.05	9.89	
Abusive supervision	0.17**	0.05	3.16	
Age	−0.01	0.07	−0.17	
Gender	−0.05	0.10	−0.55	
Education	0.06	0.07	0.83	
Work experience	0.04	0.05	0.81	

	Effect	*SE*	LL 95% CI	UL 95% CI
Indirect effect	0.08	0.03	0.0271	0.1476

	Effect	*SE*	*z*	
Normal theory test for indirect effect	0.08**	0.03	2.78	

**Note:** ***p* < 0.01, **p* <0.05. Bootstrap sample size = 5000. CI = confident interval. LU = lower limit. UL = upper limit.

**Table 5 ijerph-17-01240-t005:** Moderation effect.

Outcome: Knowledge Hiding	β	SE	*t*	R^2^
				0.13
Constant	3.46**	0.29	12.12	
Abusive supervision	0.20**	0.07	3.01	
Psychological ownership	−0.30**	0.09	−3.34	
Abusive supervision x psychological ownership	−0.24**	0.09	−2.53	
Age	0.06	0.07	0.79	
Gender	0.01	0.11	0.16	
Education	0.04	0.08	0.31	
Work experience	0.01	0.05	0.30	

**Note:** ***p* < 0.01 Bootstrap sample = 5000.

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
