# Peer review of "Tit for Tat: Abusive Supervision and Knowledge Hiding-The Role of Psychological Contract Breach and Psychological Ownership"

_ijerph, 2020, doi:10.3390/ijerph17041240_

Round 1
Reviewer 1 Report
At its current state the paper is well organized. The paper has some contribution to the field, but needs further development, especially on what concerns theoretical support. Some comments and suggestions to the authors:
The paper has some merit and might bring some contribution to the field. However, the paper must improve the literature review. The article misses here some important references from abusive supervision and knowledge hiding and well as some from psychological contract breach and psychological ownership that is so common in the literature. Some of these paragraphs are not supported. The main purpose of this study was to investigate (a) the effects of two abusive supervision and knowledge hiding used and moderating role of psychological ownership; and (b) mediating role of psychological contract breach. Also, your hypothesis need further support and the rationale behind should also be improved. In hypotheses 1 and 2 that construct is only explained after the formulation of those hypotheses and in a very weak way. Please improve this.
Research method is well organized. However, how were the constructs in analysis measured?
The article should also improve the research results discussion and contrast your results with literature and previous studies. You must also make implication for practice and theory as well as point out your study limitations.
Author Response
Reviewer 1 Comments
Comment 1: The paper has some merit and might bring some contribution to the field. However, the paper must improve the literature review. The article misses here some important references from abusive supervision and knowledge hiding and well as some from psychological contract breach and psychological ownership that is so common in the literature. Some of these paragraphs are not supported. The main purpose of this study was to investigate (a) the effects of two abusive supervision and knowledge hiding used and moderating role of psychological ownership; and (b) mediating role of psychological contract breach. Also, your hypothesis need further support and the rationale behind should also be improved. In hypotheses 1 and 2 that construct is only explained after the formulation of those hypotheses and in a very weak way. Please improve this.
Response to Comment 1
Thank you to the reviewer for suggestions.
In response to the reviewer comments, first, we have incorporated the some important and recent references from the literature of abusive supervision, knowledge hiding, psychological contract breach and psychological ownership such as Bogilović, S., Černe, M., & Škerlavaj, M. (2017). Hiding behind a mask? Cultural intelligence, knowledge hiding, and individual and team creativity. European Journal of Work and Organizational Psychology, 26(5), 710-723. De Geofroy, Z., & Evans, M. M. (2017). Are emotionally intelligent employees less likely to hide their knowledge?. Knowledge and Process Management, 24(2), 81-95. Kim, S.L., Lee, S. and Yun, S. (2016), “Abusive supervision, knowledge sharing, and individual factors: a conservation-of-resources perspective”, Journal of Managerial Psychology, Vol. 31 No. 6. Zhu, Y., Chen, T., Wang, M., Jin, Y., & Wang, Y. (2019). Rivals or allies: How performance‐prove goal orientation influences knowledge hiding. Journal of Organizational Behavior, 40(7), 849-868. Malik, O. F., Shahzad, A., Raziq, M. M., Khan, M. M., Yusaf, S., & Khan, A. (2019). Perceptions of organizational politics, knowledge hiding, and employee creativity: The moderating role of professional commitment. Personality and Individual Differences, 142, 232-237. Etc.
Second, we have revised hypotheses 1 and 2, and tried our best by explaining in a more logical way.
In text changes
“Abusive behaviors of supervisors include mocking, rudeness, public ridicule, belittlement, and other hostile behaviors [69], such as employing the silent treatment, breaking promises [30], and aggressive eye contact [78]. Past studies have shown that abusive behaviors of supervisors are associated with employees’ undesirable attitudinal and behavioral outcomes, such as low job performance [30], decrease in organizational citizenship behavior [6], increase in deviant workplace behavior, job burnout, and emotional exhaustion [1,41,42,43,64]. Moreover, prior studies tested several individual (i.e., professional commitment, psychological entitlement) and organizational factors (i.e., organizational culture, policies, leadership styles) as an antecedents with KH (e.g. [5,37, 2,22,93,50,59,65,80]). However, one such undesirable behavioral outcome is KH, which is largely ignored in the abusive supervision literature.
Knowledge hiding is the intentional attempt of an individual to conceal knowledge that has been requested by another individual [15]. KH behaviors consisted of three dimensions, namely, rationalized hiding, playing dumb, and evasive hiding [15]. In rationalized hiding, an individual is trying to give justification to the knowledge seeker or blaming a third party for not providing the requested knowledge. Similarly, in playing dumb, an individual expresses himself as ignorant of the knowledge being requested by knowledge seeker. Evasive hiding is defined as the hider providing inappropriate facts or promising to provide information in the future to mislead the knowledge seeker with no such real intentions.
Based on the social exchange theory (SET), social relationships are based on reciprocal benefits [10,17]. Consistent with the norm of reciprocity [24], the attitudes and behaviors of subordinates are linked with supervisors’ behaviors. When subordinates receive respectful treatment from the supervisors, they tend to return that with positive attitudes and behaviors. In contrast, when subordinates are treated in an abusive fashion, they are inclined to reciprocate with counterproductive behaviors [77]. In a similar vein, displaced aggression theory (DAT) [19] also suggests that individuals victimized by abusive leaders do not express their negative attitude and behaviors directly towards their supervisors because of power and authority [76]. Instead, they show their aggression towards convenient targets or victims, such as their colleagues or coworkers [45,68]. One such retaliatory reaction directed towards colleagues/coworkers may be KH behavior. On the basis of aforementioned literature and theoretical reasoning, it is expected that abusive supervision may be positively associated with KH behavior. Therefore, we hypothesize that: (H1)” (Line: 86-115)
“Psychological contract is defined as “individual beliefs, shaped by the organization, regarding terms of an exchange agreement between individuals and their organization” [63]. PCB has been defined as the cognitive perception that an employee has not received everything that was formally or informally promised by the organization (Morrison & Robinson, 1997). PCB occurs when the organization or its representatives, such as supervisors, fail to meet the expectations of their employees [47,60]. Abusive supervisors often mock employees, belittle and degrade them in front of others, making a joke of them and display other hostile behaviors, which makes subordinates feel disrespected and their expectations unfulfilled. Thus, subordinates under abusive supervisors tend to experience PCB.
PCB is related to a range of undesirable employee attitudes and behaviors. For example, psychological contract breach is negatively associated with an employee’s trust in management [81], job satisfaction [82], intentions to stay with the organization [84], employee performance [85], citizenship behaviors [84], civic virtue behaviors [86], and employee commitment [82,83]. Furthermore, some other scholars concluded that PCB is positively associated with workplace deviant behaviors [88], neglect of job responsibilities [90], job burnout [87], employee cynicism [91], higher absenteeism [81], and revenge cognitions [88].
The relationship between supervisory abuse, PCB, and knowledge hiding can be explained through SET lens.. The notion of negative reciprocity is based on SET, which suggests that when personnel perceives supervisory abuse, they are more likely to experience the breach of psychological contract, and consequently, they are motivated to react in a negative way [14]. Hence abused employees will likely experience a breach of psychological contract and consequently be motivated to exhibit counterproductive work behavior (e.g., KH behavior) in a deliberate way to balance the exchange relationship. Based on the above literature and theoretical argumentation we presume the following hypothesis: (H2)” (Line:118-141)
Reviewer 2 Report
It would be interesting to know the language in which the scales were applied. Is it in Chinese or English language?
Author Response
Reviewer 2 Comments
Comment 1: It would be interesting to know the language in which the scales were applied. Is it in Chinese or English language?
Response to Comment 1
Thank you
The scale used in Chinese language as data were collected data in the Chinese context and from Chinese respondents.
In text changes
“The scale used in the study was in Chinese language, because we collected data in the Chinese context and from Chinese respondents. For this purpose, the questionnaire was translated into Chinese through the back-translation method, recommended by Brislin [12]” (Line:183-186)
Reviewer 3 Report
Review of “Tit for Tat”
This paper addresses a very timely issue – what are some of the consequences of abusive supervision and is one of those consequences knowledge withholding? The author(s) posit a relatively simple, but elegant, model suggesting that the relationship between supervisory abuse and knowledge withholding is mediated by feelings of psychological contract breach and moderated by psychological ownership (i.e. greater psychological ownership produces more tolerance for supervisory misbehavior, etc.).
The literature review and theoretical set-up are right on-point and generally well written. The research design is, in this area of research, almost unsurpassed – the data collection over three points in time measured the three different constructs of interest (supervisory abuse at T1, psychological breach and T2, and knowledge withholding at T3). Measures were vigorously assessed for reliability and validity using CFA. And the analysis goes right to the heart of testing the hypotheses. The conclusion is (if anything) modest but on-target. In short, there’s not much not to like here and the author(s) are to be commended for their rigorous research.
I really only have one question that could be cleared up pretty quickly. How to people end up in the executive education program where the researchers get their respondents from? (I may have missed this). This is important because the selection into the program may produce people who are unusually astute at evaluating organizational dynamics, people who’ve had unusually bad experiences with supervisors (or unusually good ones), or people who may find unusual strategic value in withholding important information.
I think this minor problem could be cleared up by answering two questions that won’t take much time or text:
How do people end up in the executive education program? Do companies send them? Do people volunteer in a more-or-less open enrollment scenario? And then Is there any reason to believe the people in the program are unusually successful or have experienced unusual levels of conflict with supervisors? Are they viewed as “promising” or “troubling”? etc.
This is really the only question I have. Otherwise this is a very sound and insightful piece of research.
Author Response
Reviewer 3 Comments
Comment 1: This paper addresses a very timely issue – what are some of the consequences of abusive supervision and is one of those consequences knowledge withholding? The author(s) posit a relatively simple, but elegant, model suggesting that the relationship between supervisory abuse and knowledge withholding is mediated by feelings of psychological contract breach and moderated by psychological ownership (i.e. greater psychological ownership produces more tolerance for supervisory misbehavior, etc.).
Thank you very much
Comment 2: The literature review and theoretical set-up are right on-point and generally well written. The research design is, in this area of research, almost unsurpassed – the data collection over three points in time measured the three different constructs of interest (supervisory abuse at T1, psychological breach and T2, and knowledge withholding at T3). Measures were vigorously assessed for reliability and validity using CFA. And the analysis goes right to the heart of testing the hypotheses. The conclusion is (if anything) modest but on-target. In short, there’s not much not to like here and the author(s) are to be commended for their rigorous research.
Thank you very much for the encouragement.
Comment 3: I really only have one question that could be cleared up pretty quickly. How to people end up in the executive education program where the researchers get their respondents from? (I may have missed this). This is important because the selection into the program may produce people who are unusually astute at evaluating organizational dynamics, people who’ve had unusually bad experiences with supervisors (or unusually good ones), or people who may find unusual strategic value in withholding important information.
I think this minor problem could be cleared up by answering two questions that won’t take much time or text:
How do people end up in the executive education program? Do companies send them? Do people volunteer in a more-or-less open enrollment scenario? And then Is there any reason to believe the people in the program are unusually successful or have experienced unusual levels of conflict with supervisors? Are they viewed as “promising” or “troubling”? etc.
This is really the only question I have. Otherwise this is a very sound and insightful piece of research.
Response to Comment 3
Thank you
Our respondents consisted of full-time employees and were enrolled (only weekend days) in two large universities in China. The respondents were send by their organization as a part of their career development program.
In response to your question, “Is there any reason to believe the people in the program are unusually successful or have experienced unusual levels of conflict with supervisors?” Yes, because we only disseminated questionnaires to all those respondents, who had a very frequent interaction with their supervisors and at least one year of working experience in their respective organization. All the individuals took participation voluntarily in our study.
In text changes
“In this study, data were gathered from full-time employees enrolled (only weekend days) in an executive development program at a large universities in China. The respondents were send by their organization as a part of their career development program. A diverse pool of 344 respondents was included in the manufacturing and services sectors. According to Highhouse [33], the diversified sample of different functional groups and organizations increase the generalizability of the research findings. Our sample were consisted of those respondents who had at least one year working experience with their immediate boss in their organization. All the respondents were very promising to participate in the survey.” (176-183)